# CRABP1 in Non-Canonical Activities of Retinoic Acid in Health and Diseases

**DOI:** 10.3390/nu14071528

**Published:** 2022-04-06

**Authors:** Jennifer Nhieu, Yu-Lung Lin, Li-Na Wei

**Affiliations:** Department of Pharmacology, University of Minnesota, Minneapolis, MN 55455, USA; nhieu001@umn.edu (J.N.); yllin@umn.edu (Y.-L.L.)

**Keywords:** CRABP1, retinoic acid, neurodegeneration, inflammation, metabolism, cancer, human disease, non-canonical, MAPK, CAMKII

## Abstract

In this review, we discuss the emerging role of Cellular Retinoic Acid Binding Protein 1 (CRABP1) as a mediator of non-canonical activities of retinoic acid (RA) and relevance to human diseases. We first discuss the role of CRABP1 in regulating MAPK activities and its implication in stem cell proliferation, cancers, adipocyte health, and neuro-immune regulation. We then discuss an additional role of CRABP1 in regulating CaMKII activities, and its implication in heart and motor neuron diseases. Through molecular and genetic studies of *Crabp1* knockout (CKO) mouse and culture models, it is established that CRABP1 forms complexes with specific signaling molecules to function as RA-regulated signalsomes in a cell context-dependent manner. Gene expression data and *CRABP1* gene single nucleotide polymorphisms (SNPs) of human cancer, neurodegeneration, and immune disease patients implicate the potential association of abnormality in CRABP1 with human diseases. Finally, therapeutic strategies for managing certain human diseases by targeting CRABP1 are discussed.

## 1. Introduction: Canonical and Non-Canonical Activities of All-Trans Retinoic Acid (atRA)

Vitamin A (also known as retinol) is an essential nutrient required for almost all physiological processes [1]. The profound effects of vitamin A are elicited mainly through atRA, as well as its various isomers. Through decades of studies, it has been established that atRA, as the principal active metabolite of vitamin A, executes its activities through binding to its nuclear receptors, RA receptors (RARs), which usually pair with Retinoid X Receptors (RXRs), can bind the cis isomers of atRA. These RAR/RXR pairs, in various combinations, act to regulate the transcription of numerous target genes that harbor RA response elements (RAREs) in their regulatory regions [2,3]. RAR/RXR pairs often also act together with other transcription factors to confer further specificity in the expression of target genes, resulting in the tight regulation of specific cellular processes such as proliferation [4], differentiation [5], apoptosis [6], and other physiological functions. These ultimately ensure the homeostasis of most organ systems/physiological processes [2,7]. Dysregulated RA signaling often leads to disease conditions [8,9,10]. These RAR-mediated activities of RA, which occur in the nucleus to regulate the execution of genetic programs, generally span an extended period of time (days to years) and are referred to as canonical activities of RA [11]. An extensive body of work has determined that the cellular retinoid-binding proteins (CRABPs) I and II facilitate these canonical activities of RA. Rigorous biochemical studies have characterized classical CRABP1 functions in RA binding, sequestration, and metabolism via cytochrome (CYP) P-450 enzymes [12,13], (reviewed in-depth in [14,15,16,17]) while CRABP2 is responsible for the transport of RA to the nucleus [14,15,18,19].

In 2008, a study first reported a novel activity (effect) of atRA that occurred rapidly (within minutes) to alter the protein phosphorylation status of a transcription factor TR2 in maintaining stem cell proliferation and stemness potential [20]. Subsequent studies [21,22,23] further documented similar activities of atRA that shared several features: (1) RAR-independence, (2) occurring in the cytosol without altering gene expression, and (3) rapid (typically within minutes) action. These novel activities of atRA were collectedly referred to as “non-canonical” and were later found to be mediated by the Cellular Retinoic Acid Binding Protein 1 (CRABP1) [22]. These CRABP1-mediated non-canonical activities of atRA were ultimately validated in careful studies of *Crabp1* knockout (CKO) mice and cultures, which also revealed the physiological/disease relevance of CRABP1 [24,25,26,27,28,29,30,31,32,33]. 

Extensive molecular and cell biological studies have identified specific cytosolic signaling pathways that can be targeted by CRABP1 in a cell context-dependent manner. It is believed that CRABP1 functions as a signal integrator by forming various specific RA-regulated signaling protein complexes (signalsomes) in different cells to modulate specific cellular processes/functions. Below we summarize two validated CRABP1-mediated, non-canonical RA signaling pathways, and discuss evidence/implications for the role of CRABP1-signalsome in human diseases.

## 2. CRABP1-Signalsomes

CRABP1 is the most highly conserved retinoid-binding protein among all the known binding proteins and nuclear receptors for retinoids. CRABP1 binds, specifically, atRA with a high affinity (<1 nM) [34,35,36,37]. Given its high affinity toward atRA and cytosolic distribution, CRABP1 has been proposed and shown to sequester the poorly soluble RA from the aqueous cytosolic environment [12,13,14,15,16,17,30]. This led to the belief that CRABP1 would function to control RA availability in the cell, which indeed was supported by several molecular studies, by altering the expression level of CRABP1, that documented subsequent changes in the expression of RA-responding genes [38,39]. As introduced earlier, CRABP1 could participate in RA metabolism by delivering RA to CYP P-450 metabolic enzymes and microsomes via protein-protein interactions and substrate channeling [15,40]. However, the physiological role of CRABP1 in mediating the newly observed, non-canonical activity has remained largely elusive. Only recently, studies of CKO mice and cultures in various physiological/pathological conditions (see the following section) began to shed light on multiple functional roles of CRABP1 in modulating specific cellular processes, which contributed to the “non-canonical” activity. The fact that CRABP1 is important for multiple signaling pathways is consistent with the extremely high conservation of its amino acid sequence across animal species. Figure 1 shows the reported amino acid sequence alignment of CRABP1 among five animal species including human [41], pig [42], rat [43], mouse [44], and bovine [45]. Importantly, there is only a single residue, at position 86, that is not conserved, with alanine in human and pig sequences and proline in mouse, rat, and bovine sequences (Figure 1). 

The extreme conservation of CRABP1 during evolution would suggest important functional constraints. The evidence for this notion was obtained in careful studies of CKO mouse phenotypes (see later). Mechanistic details were provided in biochemical and cellular studies that first revealed specific context-dependent “binding partners” of CRABP1, which were rigorously defined according to at least two criteria: (a) specific and direct binding to CRABP1, which could be validated in vitro, and (b) forming specific cytosolic protein complexes that could be validated in vivo. Functional consequences of these CRABP1-containing protein complexes were each found to be capable of modulating certain specific cytosolic signaling pathways in a particular cell type. These CRABP1-containing protein complexes are therefore referred to as CRABP1-signalsomes. Currently, two types of CRABP1-signalsomes have been identified, which are discussed in the following sections.

### 2.1. CRABP1-MAPK (RAF-MEK-ERK) Signalsome in Stem Cells, Cancers

A specific Crabp1-signaling complex was first proposed after studying embryonal carcinoma (EC) and embryonic stem (ES) cells that were stimulated by a physiological concentration (10 nM) of atRA to modulate their proliferation/differentiation (reviewed in [46,47]). The initial study detected a very rapid (within minutes) response of these cells to atRA administration, which occurred in the cytosol and involved a mitogen-activated protein kinase (MAPK) pathway to modify target proteins for specific post-translational modifications [20,21,22,23]. This atRA-elicited signal was found to involve CRABP1, and could rapidly (within minutes) alter (dampen) the activity of the initiating kinase of the MAPK pathway, which is the rapidly Accelerated Fibrosarcoma (RAF) kinase and is a cell membrane-anchored kinase activated by the mitogenic signal Ras GTPase [48]. The MAPK kinase signaling cascade is comprised of Ras GTPase which activates RAF, then mitogen-activated protein kinase kinase (MEK), and then extracellular-signal-regulated kinase (ERK). Activation of this signaling pathway generally leads to cell proliferation and growth for stem/progenitor cells [48]. Through biochemical and molecular studies, it is now established that CRABP1 competes with Ras by directly interacting with RAF at its Ras-binding domain, thereby dampening MAPK signal propagation and ultimately modulating (reducing) cell proliferation of ES, EC, and neural stem cell (NSC) [22,25,29]. The proposed mechanistic model for CRABP1-MAPK signalsome is shown in Figure 2.

To this end, the physiological/pathological relevance of CRABP1 is most evident in cancers. For instance, the CRABP1 gene has been reported as a tumor suppressor or an oncogene in animals and humans [15,49,50,51,52,53,54,55,56,57,58,59,60,61]. In comparing CKO and wild-type ESCs, as well as in gain- and loss-of-functional studies of cancer cell models, it was found that CRABP1 was involved in modulating cell cycle control [22]. By competing with Ras for forming complexes with RAF/MEK, atRA-CRABP1 dampened mitogen-activated ERK activity and suppressed cell cycle progression by expanding the G1 phase [22,29]. This supports the notion that CRABP1 can be a tumor suppressor. Additional evidence supporting a functional role for CRABP1 in stem cell proliferation was obtained from studying CKO mice that were found to have expanded NSC pools (as a result of enhanced NSC proliferation in CRABP1-deleted hippocampus), which was consistent with the CKO mouse behavior indicating improved memory function [25]. Importantly, the hippocampus is among the tissues where CRABP1 is most highly expressed, especially in the NSC-rich region of the dentate gyrus. Thus, CRABP1 can participate in the homeostatic control of the NSC pool in the brain. Readers are referred to an in-depth review of this CRABP1-regulated signaling pathway by Nagpal and Wei [62].

### 2.2. Crabp1-MAPK Signalsome in Metabolism and Immunity

Lin et al. first observed that CKO mice exhibited increased high-fat diet (HFD)-induced obesity and insulin resistance (IR), suggesting a protective role for CRABP1 against the development of metabolic disorders. A molecular study of CKO mice elucidated an underlying mechanism for this metabolic phenotype that, in normal adipocytes, CRABP1 could negatively regulate ERK activity to inhibit adipogenesis and adipose hypertrophy [28]. Therefore, CKO mice are more prone to HFD-induced obesity and IR. To this end, it has been reported that pharmacological doses of RA could inhibit adipogenesis and protect against obesity, and this was attributed, primarily, to RAR-mediated activities [63,64,65,66,67]. These recent studies of CKO models revealed CRABP1 as an additional player in mediating physiological activities of atRA regarding metabolic homeostasis and the maintenance of healthy adipose tissue [28].

In examining the systemic inflammatory status/potential of CKO mice, it was found that HFD-fed CKO mice all had increased systemic inflammation, indicated by invading immune cells in adipose tissue [28], increased inflammatory driver Receptor Interacting Protein 140 (RIP140) (gene name Nrip1) [68] in the blood [31], elevation in inflammatory cytokines, and significantly enhanced macrophage M1 polarization (unpublished). Previous studies also indicated that CKO mice had overall increased inflammation in the heart, indicated by increased cardiac fibrosis [26], and an altered anxiety and stress response in their HPA axis [32]. To this end, CRABP1 was found to be involved in exosome secretion from CRABP1-expressing neurons. Specifically, the RIP140-containing exosome population was significantly expanded in the blood and cerebral spinal fluid (CSF) of CKO mice, due to, in part, increased exosome secretion from CKO neurons [31]. Importantly, these neuron-derived RIP140-containing exosomes could be engulfed by macrophages to increase their inflammatory M1 polarization, thereby increasing systemic inflammation. This study, by monitoring the intercellular transfer of the inflammatory driver, RIP140, demonstrates exosome secretion as a potent means to transfer neuronal inflammation into systemic inflammation; mechanistically, this study identifies CRABP1 as an important regulator of exosome secretion from specific CRABP1-expressing neurons, which also involves the MAPK-ERK signaling in these neurons [31]. 

### 2.3. CRABP1-CaMKII Signalsome in Cardiomyocytes and Motor Neurons (MNs)

A different CRABP1-signaling complex was identified from studying deteriorated heart function of CKO mice [26,27], and their premature weakening in motor function [33]. The expression study confirmed CRABP1 expression in cardiomyocytes [26] (relevant to the CKO heart phenotype) and motor neurons (relevant to the CKO motor function phenotype) [33]. This signaling complex is comprised of CRABP1 and calcium-calmodulin-dependent kinase 2 (CaMKII), an enzyme critical to calcium signaling/handling and highly enriched in both cardiomyocytes [69] and neurons [70,71]. It is known that CaMKII regulates contraction in cardiomyocytes [69] and long-term potentiation in neurons [70,71], respectively. Both types of cells are highly dependent upon calcium homeostasis for their functions where CaMKII is a key mediator of calcium signaling [72]. All the CaMKII isoforms have a conserved architecture comprised of the kinase, regulatory, and association/oligomerization domains, and share the same activation mechanism through the binding of calmodulin to the calmodulin-binding domain (CaMBD) within its regulatory domain. CaMKII activation occurs when intracellular (Ca^2+^) increases and binds calmodulin. Ca^2+^-calmodulin then binds and activates CaMKII, which is often marked by phosphorylation at threonine 286/7 (T286/7), depending on the CaMKII isoform [73,74]. In vitro data showed that CRABP1competes with calmodulin by directly interacting with CaMKII at the CaMBD [26,27]. Therefore, CRABP1 could dampen Ca^2+^/Calmodulin activated CaMKII activity. Since over-activation of CaMKII is a major trigger of the death/damage of cardiomyocytes [75] and neurons [76], by dampening CaMKII over-activation, CRABP1 can play a protective role in maintaining the health of both the heart and neurons. These are elaborated on in the following section. The proposed mechanistic model for CRABP1-CaMKII signalsome is shown in Figure 3. 

#### 2.3.1. CRABP1-CaMKII Signalsome in Cardiomyocytes

CKO mice naturally and gradually exhibited cardiac hypertrophy, reflected in their significantly depressed heart function in older animals [26]. Using the isoproterenol (ISO)-induced cardiomyopathy model for heart failure [77,78], studies showed that CKO mice were more sensitive/vulnerable to ISO treatment. Acute, high-dose ISO treatment activates beta-adrenergic receptors to induce acute cardiomyocyte contractions, triggering a pathological condition of heart overactivation. Chronic ISO treatment induces more severe cardiac hypertrophy, and, eventually, fibrosis and necrosis occur, mimicking heart failure in human patients [77,78]. Interestingly, in the acute ISO treated model, CKO mice were more sensitive and exhibited a significantly increased CaMKII activity marked by elevated T286 phospho-status and phosphorylation of the CaMKII cardiac substrate, PLN. These were supported by molecular studies described above, that CRABP1 dampened CaMKII activation by competing with calmodulin for its binding to CaMKII [26]. In a subsequent study [19], it was found that pretreatment with atRA before chronic ISO administration could attenuate ISO-induced heart damage and CaMKII activity in the wild type, but not the CKO mice [27]. These studies clearly demonstrated a protective role for CRABP1, as well as the potential application of CRABP1-ligand such as RA, in certain heart damage/diseased conditions.

#### 2.3.2. Crabp1-CaMKII Signalsome in MNs

CRABP1 expression is tissue and cell-type specific. In the central nervous system, it is specifically and highly expressed in spinal cord MNs [33]. These neurons project to and innervate, primarily, muscles to form tightly regulated structures called neuromuscular junctions (NMJs) [79]. Neuronal activity from MNs is propagated through NMJs to elicit muscle contraction, and calcium signaling/handling (mediated by CaMKII) is critical to the function of both presynaptic (MN) [80,81,82] and post-synaptic (muscle) [83,84] compartments. MNs release neurotransmitters such as acetylcholine to induce muscle contraction, and express Agrin, a proteoglycan essential for NMJ development and maintenance [79,85]. Crabp1 is specifically expressed in the presynaptic compartment, comprised of MNs, but is not expressed in the post-synaptic muscle compartment [33]. This study identified CRABP1-CaMKII signaling in MNs, which contributed to the regulation of Agrin expression and its downstream target, the muscular LRP4-MuSK signaling that maintained AChR clusters and healthy NMJ [79,86]. By comparing to wild-type mice, CKO mice were found to exhibit age-dependent more profound motor deterioration, reflected in their significantly reduced grip strength compared to the age-controlled group. Detailed histological studies revealed more severely damaged NMJs in CKO mice as compared to wild-type mice, characterized by irregular NMJ morphology, fragmentation, and reduced number. Consistently, in the CKO spinal cord tissues, CaMKII activity was significantly increased as compared to WT spinal tissues. Pathological CaMKII activation (overactivation) occurs in multiple disease states of the nervous system, frequently referred to as excitotoxicity [76]. In MN1 culture (a spinal MN cell line), inhibiting CaMKII via KN-93 (mimicking CRABP1 dampening effect) increased their Agrin expression, consistent with the reduction in Agrin detected in CKO NMJ tissues. It is concluded that CRABP1, in MNs, can target CaMKII to dampen its over-activation, which provides a protective mechanism against over-activation of CaMKII that could lead to MN degeneration. Importantly, re-expressing CRABP1 in young (before disease onset) CKO mice could partially rescue their motor deficits and correct CaMKII activity and Agrin expression.

## 3. Crabp1 in Two Common Human Diseases: Cancer and Neurodegeneration

CRABP1 has been studied mostly in the context of nutrition, in particular vitamin A metabolism and homeostasis. The increasingly reported biological functions of CRABP1 as described above are all very different from the canonical RAR-mediated effects that typically alter genome programming and gene expression. The physiological relevance of these CRABP1-mediated effects has been illustrated in both CKO mice and tissue culture systems which model various human diseases. The multiple functions of CRABP1 would predict numerous disease conditions where CRABP1 can be involved. Indeed, CKO mice exhibited multiple phenotypes mimicking human diseases [24,25,26,27,28,29,30,31,32]. In tissue cultures, it is possible to examine the effects of its best-known ligand, atRA, in eliciting non-canonical activities through CRABP1, and to demonstrate holo- and apo-CRABP1’s functions in specific cell types. In a genetically manipulated mouse model such as CKO, it is possible to illustrate how CRABP1 can participate in physiological processes and prevent diseased conditions/progression. However, given the technical difficulty in manipulating vitamin A and RA status in mice, the contribution of endogenous RA to the prevention of diseases, via CRABP1, remains elusive. Nevertheless, the implication of CRABP1 in human diseases can be uncovered by mining the available human data sets and literature, which has yielded some interesting information supporting a potential role for CRABP1 in human diseases. Below, we discuss several human studies/data sets that have revealed altered expression or protein sequence of CRABP1 in human patients. First, the reported genetic association of CRABP1 in various human diseases is summarized in Table 1, followed by a discussion on specific implications in cancers, neurodegeneration, and other rare diseases. The relevant accession IDs of CRABP1 expression studies from the EMBL-EBI Expression Atlas Data Repository [87] are provided in Table 1.

### 3.1. CRABP1 in Cancers

Dysregulation of CRABP1 expression in cancers is a well-documented phenomenon (Table 1; [15,49,50,51,52,53,54,55,56,57,58,59,60,61]). Furthermore, cancer databases such as The Cancer Genome Atlas (TCGA) and cBioPortal [99,100] for Cancer Genomics have revealed numerous single nucleotide polymorphisms (SNPs) in patients across various cancers. These SNPs could result in various defects in CRABP1 such as synonymous mutation, splicing alternation, missense mutation, and augmented expression levels. Figure 4a lists SNPs present in patients from various cancer types that occurred in the −3 kb upstream regulatory region, which could affect *CRABP1* expression levels; Figure 4b lists SNPs in the coding region that could alter the CRABP1 sequence. However, no experimental data have been provided to validate the “disease association” of these SNPs. Nevertheless, given the conservation of CRABP1 across mammals, any alterations in CRABP1 caused by these SNPs could potentially disturb CRABP1 functions and normal cellular processes especially proliferation which could impact tumor formation or progression.

### 3.2. CRABP1 in Neurodegeneration

CRABP1 expression has been found to be reduced in the following neurodegenerative disease conditions: amyotrophic lateral sclerosis, spinal muscular dystrophy, and age-related macular degeneration. Data mining of the ALS Variant Server (http://als.umassmed.edu/, accessed on 31 January 2022) revealed several SNPs present in ALS patients that are located in the upstream regulatory region or in the coding region of CRABP1 (Figure 5a,b). These SNPs in ALS patients could potentially alter CRABP1 levels or functions, thereby contributing to disease initiation or progression. However, experiments are needed to verify the disease relevance of these SNPs. Interestingly, a study by Jiang et al. identified CRABP1 as the most significantly suppressed gene in ALS patients’ motor neurons (MNs) as compared to healthy subjects, suggesting that CRABP1 may play a role in ALS etiology. This is consistent with the severe motor degeneration phenotype of CKO mice in older age groups [33]. It would be interesting to experimentally examine the potential contribution of SNPs identified from the AVS database in various neurodegenerative diseases. 

The importance of CRABP1 in neurons, particularly MNs, is further supported by the finding that the mouse Crabp1 gene is tightly regulated by sonic hedgehog (Shh) [30], a potent inducer of motor neuron differentiation [101]. It appears that Shh activates glioma-associated oncogene homolog 1 (Gli1) that binds the Gli target sequence in Crabp1′s regulatory region, thereby up-regulating Crabp1 expression [30]. Therefore, for MN differentiation and function, proper expression of CRABP1 is important.

### 3.3. CRABP1 in Rare Human Diseases

Altered CRABP1 level or function has also been observed in other diseases. In Moyamoya Disease (MMD), a vascular disease characterized by progressive occlusion of cerebral arteries [102], CRABP1 protein level was found to be increased in the CSF of MMD patients [96]. Kim et al. speculated that the increase in CRABP1 during MMD progression might disrupt the regulatory activity of retinoids on growth factor signaling responsible for arterial occlusion [96]. A study by Hur et al. also speculated an increase in CRABP1 as a potential biomarker of diabetic neuropathy [97]. CRABP1 has also been implicated in HIV therapy associated with lipodystrophy and metabolic disorder. Carr et al. proposed that the toxic effects on adipose and metabolism associated with the use of HIV-1 protease inhibitors were, in part, due to these inhibitors’ direct binding and inhibiting CRABP1 function [98]. However, no experimental data have been presented to substantiate or support a role for CRABP1 in these rare human diseases.

## 4. Conclusions and Future Directions

The CRABPs have been established as key players in RA binding, sequestration, metabolism, and nuclear transport to RARs. In addition to these classical functions, novel roles in the CRABPs have also been observed, such as CRABPII in RNA transcript stabilization [103,104] and as a tumor suppressor in breast cancer [105,106]. Here, we have reviewed the novel functions of the CRABP1 as signalsomes, particularly in the physiological contexts of (1) MAPK regulation in growth, cancer, metabolism, and immunity and (2) CaMKII regulation in cardiomyocyte and motor neuronal function. 

Clinically, RA and its analogs have long been proposed for therapeutic applications in managing different diseases [107,108]. Through decades of studies, most of these efforts have not proven to be fruitful because of the wide spectrum of retinoid toxicities. This has presented a particularly serious concern in using retinoids to manage chronic diseases such as metabolic/inflammatory/neurological diseases [109,110,111,112,113]. The most efficacious application of retinoids has been in topical application, such as for treating acne vulgaris [114] and in aggressively treating severe or end-stage cancer patients, particularly an aggressive form of leukemia (acute myeloid leukemia) [115]. Most other attempts have proven to be not successful due to toxic side-effects which are caused by the RAR/RXR-mediated activities. Given that CRABP1 is specifically expressed in limited types of cells and only in certain stages of cell differentiation, and that CRABP1 participates in very specific signaling pathways, it might be more feasible by targeting CRABP1 using selective ligands that do not act on RARs or RXRs. This strategy exploits the collected evidence, as reviewed here, that CRABP1 mediates non-canonical RA signaling pathways that are cell type- and context-specific. 

Currently, two novel CRABP1-selective compounds, C3 and C4, have been documented, which have been shown to modulate, specifically, the MAPK signaling pathway in CRABP1-expressing cells [24,31]. The efficacy of C3 and C4 has been demonstrated to induce apoptosis (in cancer cells) [24] and regulate exosome secretion (in neurons) [31]. These in vitro results would encourage further exploitation of this potential therapeutic strategy, such as in managing cancers and inflammation. Other groups have recently explored the use of synthetic ligands to target CRABP1. Tomlinson et al. determined the crystal structures of CRABP1 bound to fatty acids and a synthetic retinoid, DC645 [116]. DC645 appeared to bind CRABP1 in a manner similar to that of RA, and the binding resulted in minimal structural changes in CRABP1. Interestingly upon ligand binding, side-chains on the beta-sheet surface underwent conformational re-arrangements. Therefore, structural information obtained from these biophysical studies supports our fundamental hypothesis that CRABP1 signalsome acts, primarily, through its surface interactions that involve the beta-sheet face of CRABP1. Zheng et al. determined that Maprotiline can directly bind and inhibit CRABP1, resulting in dampened ERK-mediated SREBP2 activity and ultimately reducing tumor growth in a hepatocarcinoma xenograft model [117].

Additionally, the possibility of targeting CRABP1, such as by gene or cell therapy, is underscored by clinical data of human studies which have clearly implicated that a reduction in CRABP1 level was correlated with disease severity or its progression. To correct this deficiency, gene therapy (to deliver CRABP1-expressing vector) may be carried out to target implicated tissues. Further, a cell therapy-based strategy may also be feasible. For instance, CRABP1-expressing adipocytes may be locally delivered to adipose tissues to help correct the abnormally expanded obesity.

For future studies, the most important task would be to identify and develop CRABP1-selective and signaling pathway-specific ligands that do not elicit RAR-mediated toxicity. In addition to synthetic compounds, it would be of great interest to identify and study compounds derived from naturally occurring sources, such as plants and meats. These naturally occurring ligands, if present, would be very useful to the understanding and application of nutrients that may enhance the potential physiological and protective functions of CRABP1 signalsomes. It would also be of great interest to identify other components and networks that may comprise new CRABP1 signalsomes which remain to be uncovered. Finally, a more systemic investigation into human diseases where CRABP1 could play a role is needed. Given that CKO mice have CRABP1 deleted from birth, their disease spectrum may not reflect the entire spectrum of human diseases involving CRABP1. Human genetic association studies can provide important clues into this important research direction, and may uncover more physiologically important CRABP1-signalsomes that can also deliver non-canonical activities of RA.

## Figures and Tables

**Figure 1 nutrients-14-01528-f001:**
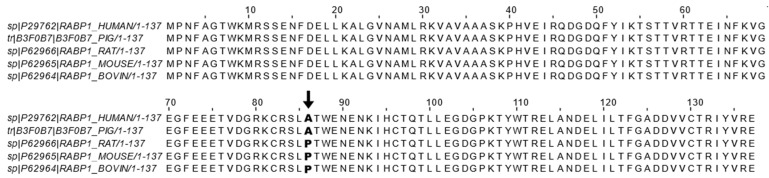
CRABP1 sequence alignment across mammals. Reported CRABP1 protein sequences from the Uniprot database of human (ID: P29762) [41], pig (ID: B3F0B7) [42], rat (ID: P62966) [43], mouse (ID: P62965) [44], and bovine (ID: P62964) [45] were aligned using the ClustalWS alignment algorithm in Jalview. Only the residue at position 86 (indicated by bold text and arrow ↓) is not conserved and exists either as an alanine (A) in human and pig sequences or as a proline (P) in bovine, rat, and mouse sequences.

**Figure 2 nutrients-14-01528-f002:**
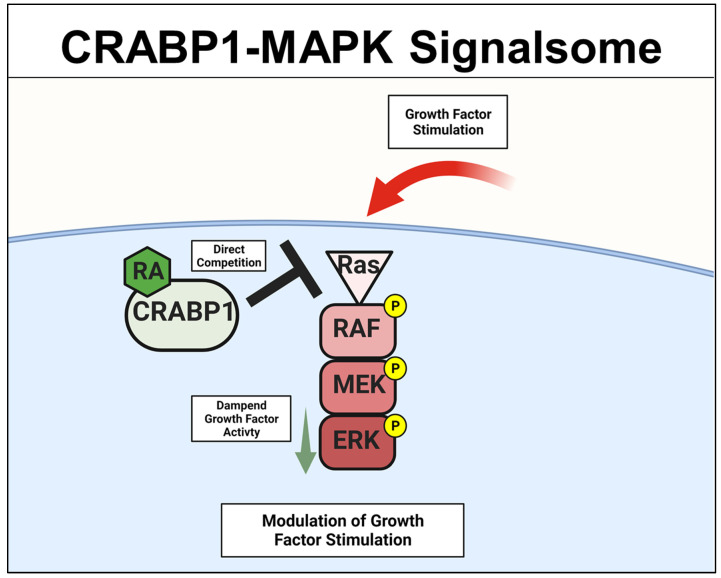
CRABP1-MAPK signalsome. The action of CRABP1-signalsome in growth-factor stimulated MAPK activity is mediated by its direct competition with Ras, resulting in dampened MAPK activation. CRABP1: Cellular Retinoic Acid Binding Protein 1, RA: retinoic acid, RAF: rapidly Accelerated Fibrosarcoma, MEK: mitogen-activated protein kinase kinase, ERK: extracellular-signal-regulated kinase.

**Figure 3 nutrients-14-01528-f003:**
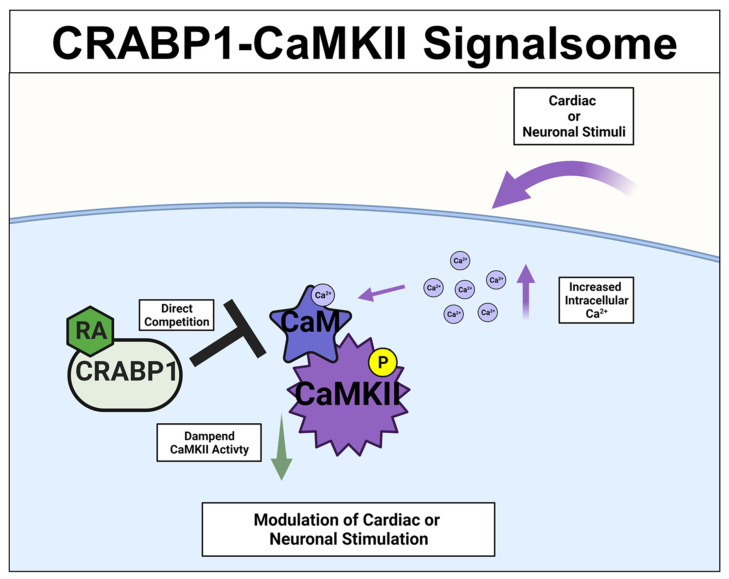
CRABP1-CaMKII signalsome. Upon cardiac or neuronal stimulation and subsequent intracellular Ca^2+^ increase to activate CaMKII, CRABP1 directly competes with calmodulin (CaM) to dampen CaMKII enzyme activity to ultimately modulate cardiac and/or neuronal stimulation. CRABP1: Cellular Retinoic Acid Binding Protein 1, RA: retinoic acid, CaMKII: calcium-calmodulin-associated dependent kinase 2.

**Figure 4 nutrients-14-01528-f004:**
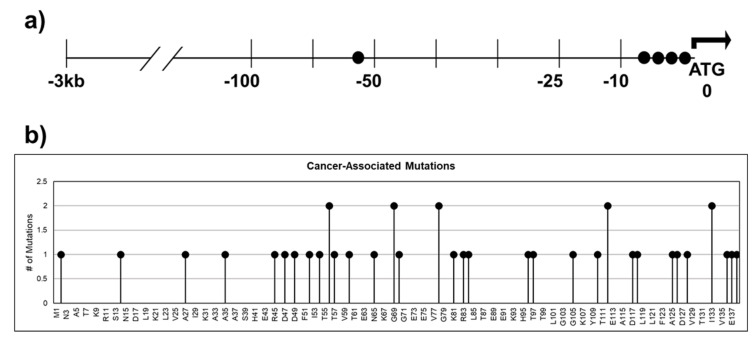
SNPs identified in cancer patients within the 3 kb upstream regulatory region and the coding sequence of CRABP1. (**a**) Gene diagram of the 3 kb upstream region of CRABP1 with SNPs denoted by “●”. (**b**) Lollipop plot indicating the location of amino acid mutations as a consequence of cancer-associated SNPs.

**Figure 5 nutrients-14-01528-f005:**
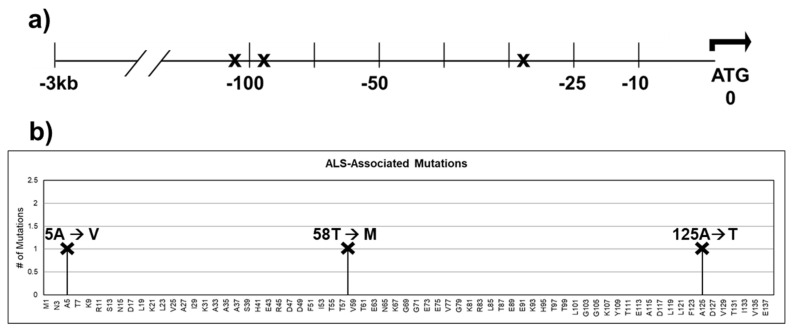
SNPs identified within the 3 kb upstream regulatory region and the coding sequence of CRABP1 in ALS patients. (**a**) Gene diagram of the 3 kb upstream region of CRABP1 with SNPs denoted by “X”. (**b**) Lollipop plot indicating the location of amino acid mutations as a consequence of ALS-associated SNPs. Altered residues are marked above each lollipop.

**Table 1 nutrients-14-01528-t001:** Changes in CRABP1 detected in human patients.

Cancer Type	CRABP1 Status	Reference
Breast Cancer	Over-Expression	[61]
Prostate Cancer	Over-Expression	[60]
Mesenchymal & Neuroendocrine Tumors	Over-Expression	[59]
Head and Neck Squamous Cell Carcinoma (HNSCC)	Over-Expression	[58]
Colorectal Cancer	Silenced (Promoter Hypermethylation)	[57]
Thyroid Cancer	Silenced (Promoter Hypermethylation)	[56]
	Reduced Expression	[54]
Ovarian Cancer	Silenced (Promoter Hypermethylation)	[53]
	Reduced Expression	[52]
Esophageal Squamous-Cell Carcinoma (ESCC)	Silenced (Promoter Hypermethylation)	[51]
Renal Cell Carcinoma	Reduced Expression	[50]
Acute myeloid leukemia (AML)	Silenced (Promoter Hypermethylation)	[49]
**Neurodegenerative Diseases**	**CRABP1 Status**	**Reference**
Amyotrophic Lateral Sclerosis (ALS)	Reduced Expression	[88]
Spinal Muscular Atrophy (SMA)	Reduced Expression	[89]
Late-Stage Age-Related Macular Degeneration (AMD)	Reduced Expression	[90]
**Immune Disorders**	**CRABP1 Status**	**Reference**
Multiple Sclerosis	Reduced Expression	[91]
Cutaneous Lupus Erythematosus (CLE)	Reduced Expression	[92] ^#^ E-MTAB-5542
Psoriasis	Reduced Expression	[93] ^#^ E-GEOD-52471
Vitiligo	Reduced Expression	[94] ^#^ E-GEOD-65127
Inflammatory Bowel Disease (IBD)	Silenced (Promoter Hypermethylation)	[95]
**Other Diseases**	**CRABP1 Status**	**Reference**
Moyamoya Disease (MMD)	Increased Protein Level	[96]
Diabetic Neuropathy	Increased Expression	[97]
HIV Therapy-Associated Lipodystrophy and Metabolic Syndrome	Inhibited Function	[98]

^#^ EMBL-EBI Expression Atlas Data Repository Accession ID.

## Data Availability

Not applicable.

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
