# Peer review of "CRABP1 in Non-Canonical Activities of Retinoic Acid in Health and Diseases"

_nutrients, 2022, doi:10.3390/nu14071528_

Round 1

Reviewer 1 Report

This is a comprehensive short review describing Canonical and Non-canonical retinoid actions. It stresses the high level of conservation of CRABP1 and the importance of the CRABP1-MAPK (RAF-MEK-ERK) signalsome in stem cells and cancer biology. CRABP1 roles in inflammation, cardiomyocyte and motoneuron survival, and cancer biology indicate potentially important roles in human disease.

Tables are useful. This is a nice update from the authors 2019 review.

Note the summary figures in the 2019 review, though, were a nice illustration of general regulatory mode of this singling pathway. Updating and including  similar  figures might improve the comprehension of this important singling network for those outside of this field. Even without these changes, this review is well done .

Author Response

Reviewer #1

Comments and Suggestions for Authors:

This is a comprehensive short review describing Canonical and Non-canonical retinoid actions. It stresses the high level of conservation of CRABP1 and the importance of the CRABP1-MAPK (RAF-MEK-ERK) signalsome in stem cells and cancer biology. CRABP1 roles in inflammation, cardiomyocyte and motoneuron survival, and cancer biology indicate potentially important roles in human disease.

Tables are useful. This is a nice update from the authors 2019 review.

Note the summary figures in the 2019 review, though, were a nice illustration of general regulatory mode of this singling pathway. Updating and including similar figures might improve the comprehension of this important singling network for those outside of this field. Even without these changes, this review is well done.

---- Response: We have added summary figures for the CRABP1-MAPK signalsome (Figure 2) and CRABP1-CaMKII signalsome (Figure 3).   

Reviewer 2 Report

Here Nhieu and colleagues presented a comprehensive review of Cellular Retinoic Acid Binding Protein 1 (CRABP1) which acts as a mediator of non-canonical activities of retinoic acid (RA), and the relevance to human diseases. 

However, there are some minor concerns to be addressed.

To begin with, the authors are highly recommended to include a schematic illustration of CRABP1-signalsomes which will help reader grasp the insights.

Further, a recent elegant work by Charles W E Tomlinson entitled “Structure-functional relationship of cellular retinoic acid-binding proteins I and II interacting with natural and synthetic ligands” may provide more pharmacological insights of CRABP1. This new line of evidence should be included in this review.

Finally, are there any natural occurring compounds that derived from foods, medical plants, or others that have CRABP1 modulation effects? These functional nutrients and their resources will promote future development of functional foods.

Collectively, a Minor Revision is recommended for this beautiful review!

Author Response

Reviewer #2

Comments and Suggestions for Authors

Here Nhieu and colleagues presented a comprehensive review of Cellular Retinoic Acid Binding Protein 1 (CRABP1) which acts as a mediator of non-canonical activities of retinoic acid (RA), and the relevance to human diseases.

However, there are some minor concerns to be addressed.

To begin with, the authors are highly recommended to include a schematic illustration of CRABP1-signalsomes which will help reader grasp the insights.

---- Response: We have added summary figures for the CRABP1-MAPK signalsome (Figure 2) and CRABP1-CaMKII signalsome(Figure 3).  

Further, a recent elegant work by Charles W E Tomlinson entitled “Structure-functional relationship of cellular retinoic acid-binding proteins I and II interacting with natural and synthetic ligands” may provide more pharmacological insights of CRABP1. This new line of evidence should be included in this review.

---- Response: Thank you. We have added text and the citation about this study from Tomlinson et al. in Section 4, paragraph 2.

Finally, are there any natural occurring compounds that derived from foods, medical plants, or others that have CRABP1 modulation effects? These functional nutrients and their resources will promote future development of functional foods.

---- Response: Natural compounds that modulate the CRABP1 signalsome are of great interest to our future study, especially from a nutritional perspective. We have discussed this future direction in Section 4 last paragraph.